# OceanGPT: A Large Language Model for Ocean Science Tasks

## Abstract

Ocean science is of great significance given that oceans cover over 70% of our planet's surface. Recently, advances in Large Language Models (LLMs) have transformed the paradigm in natural science. Despite the success in other domains, current LLMs often fall short in catering to the needs of domain experts like oceanographers, and the potential of LLMs for ocean science is under-explored. The intrinsic reasons are the immense and intricate nature of ocean data as well as the necessity for higher granularity and richness in knowledge. To alleviate these issues, we introduce **OCEANGPT**, the first-ever large language model in the ocean domain, which is expert in various ocean science tasks. We also propose DOINSTRUCT, a novel framework to automatically obtain a large volume of ocean domain instruction data, which generates instructions based on multi-agent collaboration. Additionally, we construct the first oceanography benchmark, **OCEANBENCH**, to evaluate the capabilities of LLMs in the ocean domain. Though comprehensive experiments, our **OCEANGPT** not only domontrates a higher level of knowledge expertise for oceans science tasks but also gains preliminary embodied intelligence capabilities in ocean technology.

## 1 Introduction

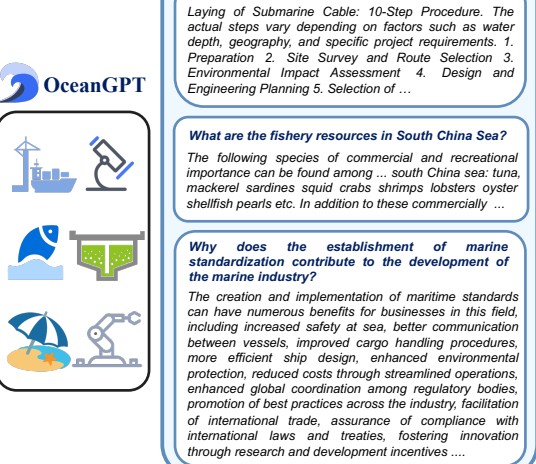

Figure 1: Capabilities of **OCEANGPT**. Our proposed model not only shows a higher level of knowledge expertise for oceans science tasks but also gains preliminary embodied intelligence capabilities in ocean technology.

Ocean science, which delves into the intricacies of oceans that cover over 70% of our planet's surface, is essential not only for understanding the rich reservoirs of life and biodiversity but also for recognizing their pivotal role in regulating the global climate and supporting economies [Esaias *et al.*, 1998; Falkowski, 2012; Visbeck, 2018; Jin *et al.*, 2023]. Recently, advances in Large Language Models (LLMs) [OpenAI, 2023; Jiang *et al.*, 2023; Zha *et al.*, 2023; Yin *et al.*, 2023; Zhao *et al.*, 2023] have

transformed the paradigm in science domains such as medical science [Moor *et al.*, 2023], molecular science [Fang *et al.*, 2023], protein science [Lin *et al.*, 2023] and geoscience [Deng *et al.*, 2023]. However, the potential for the large language model in ocean science is under-explored.

Despite remarkable success in general domain, current LLMs still do not fully meet the specific demand of oceanographers. This inadequacy is primarily due to: (1) The immense volume and intricate nature of ocean data. As ocean science research progresses, acquiring data becomes increasingly challenging, which makes enhancing the oceanic understanding both a golden opportunity and a significant hurdle. (2) The necessity for higher granularity and richness in knowledge. Note that the data requirements faced by researchers are becoming increasingly intricate and diverse. Ocean science encompasses various domains and subjects, each with its distinct data attributes and patterns.

To alleviate these issues, we introduce **OCEANGPT**, the first-ever LLM in the ocean domain, which is expert in various ocean science tasks. Specifically, we propose DOINSTRUCT, an efficient ocean science instruction generation framework that capitalizes on multi-agent collaboration. Each agent in our designed framework is considered as an expert in a specific domain (science and research, resources and development, ecology and environment etc.) and is responsible for generating the corresponding data. For the advancement of ocean science research using LLMs, we also create a benchmark called **OCEANBENCH** to evaluate the capabilities in ocean science tasks.

Through extensive experiments, **OCEANGPT** shows superiority for diverse ocean science tasks. Note that our benchmark data is based on criteria manually evaluated by ocean experts, and can accurately reflect the capabilities that LLMs possess in the field of ocean science. As depicted in Figure 1, our model can comprehensively answer questions according to the instructions of oceanographers, which demonstrates its expertise in oceanography. We further explore the potential of **OCEANGPT** from the perspectives of ocean engineering. Specifically, we integrate ocean robotics instructions into the training data and evaluate its ability via code or console commands. **OCEANGPT** not only demonstrates a higher level of knowledge expertise but also gains preliminary embodied intelligence capabilities in ocean technology.

Our contributions can be summarized as follows:

- We introduce **OCEANGPT**, the first ocean LLM, which shows superiority for various ocean science tasks. It can answer oceanographic questions according to the instructions of oceanographers, demonstrating expertise in oceanography.

- We propose DOINSTRUCT, an automated domain instruction evolving framework that constructs the ocean instruction dataset by multi-agent collaboration. Our framework effectively alleviates the difficulty of obtaining ocean domain data.

- Extensive experiments demonstrate the superiority of **OCEANGPT** in the **OCEANBENCH**. **OCEANGPT** not only demonstrates a higher level of knowledge expertise for oceans science tasks but also gains preliminary embodied intelligence capabilities.

## 2 Related Work

**Large Language Models.** The landscape of LLM [Brown *et al.*, 2020; Chowdhery *et al.*, 2022; Touvron *et al.*, 2023a,b] has rapidly evolved and achieved a series breakthroughs. Rae *et al.* [2021]; Zhang *et al.* [2022]; Thoppilan *et al.* [2022]; Scao *et al.* [2022]; Zeng *et al.* [2023] have explored the performance across a wide range of model scales and broadened the application scope [Qiao *et al.*, 2023a; Zhang *et al.*, 2023a; Qiao *et al.*, 2023b; Wang *et al.*, 2023a; Xi *et al.*, 2023]. Retrieval-Augmented Generation (RAG) is a useful solution by incorporating knowledge from external databases [Gao *et al.*, 2023; Lewis *et al.*, 2020; Schick *et al.*, 2023; Khandelwal *et al.*, 2020]. To align LLMs, instruction tuning [Wei *et al.*, 2022; Zhang *et al.*, 2023b; Ouyang *et al.*, 2022; Taori *et al.*, 2023; Wang *et al.*, 2023d; Chiang *et al.*, 2023; Xu *et al.*, 2023] is a crucial technique to alignment with user preferences and desired outputs. Different from those, we train a totally new ocean science large language model and introduce an effective domain instruction generation framework via multi-agent collaboration.

**Science Large Language Models.** LLMs have emerged as cornerstone models in addressing challenges within scientific research. Singhal *et al.* [2022] explores the potential of clinical LLMs and

introduces a human evaluation framework and instruction prompt tuning. Moor *et al.* [2023] proposes generalist medical AI that is capable of handling diverse medical tasks using self-supervised learning on large datasets. Kraljevic *et al.* [2021] introduces MedGPT, a model using EHR data and Named Entity Recognition tools for predicting future medical events. BioGPT [Luo *et al.*, 2022] is a language model pre-trained on biomedical literature for improved text generation and mining. Theodoris *et al.* [2023] describes Geneformer, a model pre-trained on single-cell transcriptomes for making predictions with limited data in network biology. Lin *et al.* [2023] demonstrates the prediction of atomic-level protein structure from primary sequences using scaled-up language models. Deng *et al.* [2023] introduces the first LLM specifically designed for geoscience, including its training and benchmarking protocols. Chen *et al.* [2023] presents tele-knowledge pre-training for fault analysis. Different from previous works, we design the first large language model for ocean science tasks and explore its potential for ocean research.

## 3   OCEANGPT

To obtain **OCEANGPT**, we firstly construct the training corpus for ocean science and pre-train an ocean LLM based on LLaMA-2 Touvron *et al.* [2023b] in Section 3.1. Then we propose DOINSTRUCT, an automated framework for domain instruction generation to build an ocean domain-specific instruction dataset. Our framework leverages multi-agent collaboration and utilizes ocean literature to automatically generate a large volume of domain-specific instructions for ocean science tasks (Section 3.2). The overview training procedure of our **OCEANGPT** is shown in Figure 2.

### 3.1   Pre-training Stage

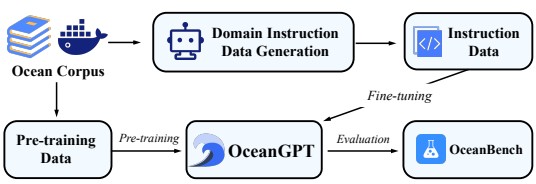

Figure 2: Overall framework of **OCEANGPT**.

To pre-train the foundation model for ocean science tasks, it is essential to construct the pre-training corpus specific to ocean science. Therefore, we firstly collect a raw corpus of 67,633 documents from **open-access literature**. For the specific volumes we choose, we prefer to consider publications from recent years to ensure the inclusion of the latest research and developments. At the same time, we select some historically significant literature to help the LLM understand the developmental history of the field. For diversity, we choose articles from different sources to ensure coverage of various research perspectives and methods. Specifically, we utilize the Python package *pdfminer* to convert the content of literature files into plain text. To ensure the quality and consistency of the data, further processing of the collected dataset is necessary. We apply regular expressions to filter out figures, tables, headers, footers, page numbers, URLs and references. Additionally, any extra spaces, line breaks, and other non-text characters are removed. The processed documents cover various aspects of ocean science such as ocean physics, ocean chemistry, ocean biology, geology, hydrology, etc. It is important to note that special characters, emoticons, and garbled characters are also replaced or eliminated during this process. We also employ *hash-based methods* to de-duplicate the data, which helps reduce the risk of over-fitting during pre-training and enhances its generalization capability.

### 3.2   Domain Instruction Data Generation

As ocean science research deepens, researchers are facing increasingly complex and diversified data demands. Ocean science corpus contains multiple fields and topics, and each topic has its unique data characteristics and patterns. To effectively simulate and obtain those data, we propose a domain instruction generation framework DOINSTRUCT to obtain ocean instructions $H$ by multi-agent collaboration. Each agent is considered as an expert in a **specific domain (topic)** and is responsible for generating the corresponding data. It not only ensures the professionalism and accuracy of the data but also allows for the parallel and efficient generation of a large amount of data. Note that the proposed framework also has greater flexibility, allowing us to independently optimize and adapt to different science domains (e.g., astronomy).

**Ocean Topic Definition.**   To provide researchers with a clear and organized resources, we manually categorize the data in ocean science into five major

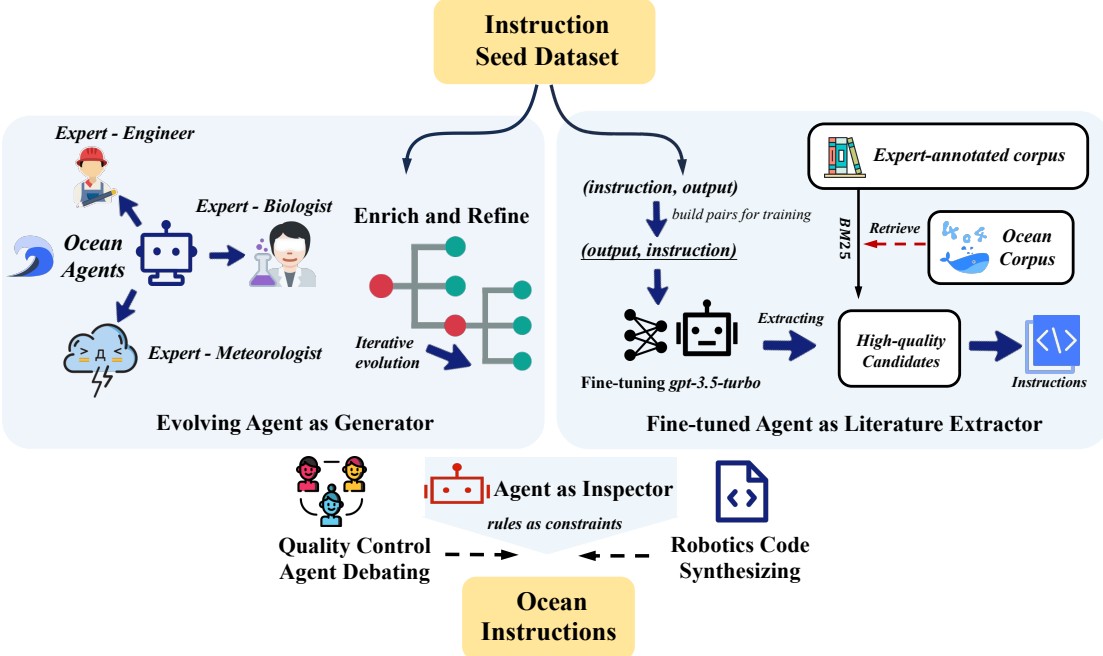

Figure 3: Procedure of our proposed DOINSTRUCT. We use agents (*gpt-3.5-turbo*) as experts for each **ocean topic** and make them rapidly expand the instructions by collaboration. In this framework, we design three agent roles: **evolving generator**, **fine-tuned literature extractor** and **inspector with rule constraints**.

ocean topics, which are based on the expertise of experts in oceanography. The definitions of these five topics comprehensively cover all the main areas of ocean science and are relatively independent. The detailed explanation for the five major topics is described as follows:

- *Science and research* focuses on the fundamental scientific theories and research related to the ocean, such as ocean currents, sea temperatures and ocean biodiversity. This portion of data separately helps drive the advancement of pure scientific research and theories.

- *Resources and development* includes fisheries, minerals, oil and gas, as well as other sustainable development resources. It is set for a better examination and planning of the rational development of ocean resources.

- *Ecology and environment.* Environmental protection and ecological sustainability are currently global hot topics. It helps to address issues such as ocean pollution, ecological degradation, and the impact of climate change on the oceans in a more focused manner.

- *Technology and engineering* encompasses aspects ranging from ocean measurements, observational equipment, and ship engineering to ocean energy development. Such categorization aids in a more focused exploration of ocean engineering and technological needs, while also facilitating interdisciplinary research with other engineering disciplines.

- *Life, culture and others.* The ocean is not only a natural resource or a subject of scientific research; it is also an integral part of culture and lifestyle. This category consists of aspects ranging from history and culture to the mutual influences between the ocean and human societal activities, such as tourism, leisure.

While these five topics are distinct, there might be some overlap as well. For instance, some issues related to ocean environmental protection might also be associated with the technology of ocean engineering. For the sake of convenience in data analysis, in the actual construction of the dataset, we map

each sample to the most relevant category.

**Agents as Domain (Ocean) Experts.** In Figure 3, we use agents as domain experts for each ocean topic and make them rapidly expand the instructions by collaboration. We collect the seed instruction data and propose three strategies by using multiple agents acting as experts.

To construct the seed dataset, we employ dozens of annotators with rich backgrounds in marine science. Each annotator is responsible for several topics and they first manually write some representative example for each marine topic. Then we use LLMs to mimic the existing data to generate a large number of similar samples. All samples are ultimately manually checked by the annotators. The entire process is very time-consuming, with all the experts spending a total of four days to validate the seed data. The final seed instruction dataset includes 5 major categories, over 500 sub-categories and a total of more than 10,000 data samples.

- *Evolving Agent as the Generator*. We design an evolving approach that selects samples from the seed dataset and simultaneously calls upon two agents (*gpt-3.5-turbo*) to evolve the selected samples. The evolution procedure includes two aspects: (1) we enrich the content of the sample by having the agent automatically add relevant background knowledge to it; (2) we guide the agent to refine the sample by conducting a more in-depth analysis of specific concepts or entities. Through multiple rounds of iterative execution, our method can rapidly expand the existing seed dataset, which allows for the rapid expansion of both the breadth and depth of information.

- *Fine-Tuned Agent as the Literature Extractor*. As shown in Figure 3, we collect a smaller expert-annotated corpus and use the *BM25* to retrieve high quality sentences in a larger ocean corpus. We regard the retrieved texts as high-quality candidate samples. Meanwhile, we fine-tune *gpt-3.5-turbo* with the seed instruction dataset, regarding the fine-tuned agent as the literature extractor. In other words, it can automatically extract instructions (*inst*) from the unannotated ocean science corpus (*output*). Therefore, we utilize the agent to automatically build pairs of *(inst, output)* on external ocean science literature.

---

**Algorithm 1** Domain Instruction Data Generation

**Require:**
Seed dataset $S$ with format $(inst, output)$,
Ocean literature corpus $O$,
Pre-defined rules $R$ for filtering
**Ensure:**
High-quality instruction dataset $H$

1: Initialize empty datasets.
$Step1Data = \emptyset, Step2Data = \emptyset, H = \emptyset$
{**Agent Collaboration as Domain Experts.**}
2: **for** each sample in $S$ **do**
3: $\quad inst, output \leftarrow sample$
4: $\quad enriched\_sample \leftarrow Enrich(inst, output)$
5: $\quad refined\_sample \leftarrow Refine(inst, output)$
6: $\quad Step1Data \quad\leftarrow\quad Step1Data \quad\cup$
$\quad enriched\_sample \cup refined\_sample$
7: **end for**
8: { **Fine-Tuned Agent as Literature Extractor.**}
9: $RetrievedTexts \leftarrow BM25\_Retrieve(O)$
10: $Model\ M \leftarrow FineTune(S_{reverse})$
11: **for** each document in $RetrievedTexts$ **do**
12: $\quad output \leftarrow document.content$
13: $\quad inst \leftarrow M(output)$
14: $\quad Step2Data \leftarrow Step2Data \cup (inst, output)$
15: **end for**
{ **Agent as Inspector with Rule Constraints.**}
16: $MergedData \qquad\qquad\qquad \leftarrow$
$Inspector(Step1Data, Step2Data, R)$
{ **Quality Control by Debating.**}
17: **for** each sample in MergedData **do**
18: $\quad inst, output \leftarrow sample$
19: $\quad debate\_result \leftarrow Debate(inst, output)$
20: $\quad$ **if** debate_result is high-quality **then**
21: $\quad\quad H \leftarrow H \cup sample$
22: $\quad$ **end if**
23: **end for**
24: **return** $H$

---

- *Agent as the Inspector with Rule Constraints*. For the massively generated instructions, we use the pre-defined rules as constraints and perform filtering on the data. These rules include syntactic and semantic constraints as well as basic definitions in the ocean domain. We describe these rules using natural language because many constraints and norms related to ocean science cannot be directly represented with expressions. Therefore, we provide prompts to the *gpt-3.5-turbo* API as demonstrations, letting it play the role of an inspector. Our method ensures that the generated

ocean instruction data is of higher quality. Detailed prompt is shown in Table 5.

Finally, we assign two extra *gpt-3.5-turbo* agents as roles to debate the quality of data and obtain high-quality instruction dataset. Our designed framework can rapidly constructing a ocean science dataset using multi-agents, and by incorporating external knowledge from marine literature, it overcomes the limitations inherent to the agents themselves. Our framework can also be effectively applied to the instruction data construction in other scientific domains. It should be noted that we separately synthesize robot instructions to equip **OCEANGPT** with the capability to interact with the environment. The procedure is in Algorithm 1 and the statistics of dataset is in Figure 4.

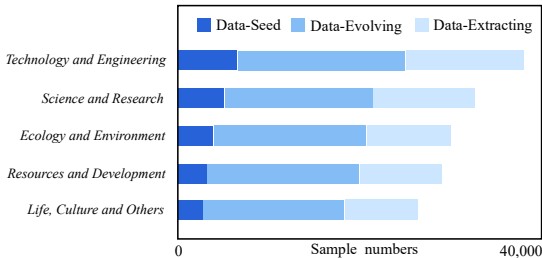

Figure 4: Statistics of our final instruction dataset. We use DOINSTRUCT to expand more than 150,000 instructions (*data-evolving*, *data-extracting*).

**Quality Control for the Dateset.** We ask domain experts to carefully review and check data to ensure quality. Specifically, the human volunteers are first trained to make sure they have a comprehensive understanding of the task. Then, we develop a platform that can help experts to randomly sample 10% instances from the generated instruction dataset. Next, the trained domain experts are asked to validate if there are potential errors in the sampled instances. The final IAA (inter-annotator agreement) score for our dataset is 0.82, which satisfies the research purpose.

## 4 Benchmarking Ocean Science Tasks

We provide detailed explanations of the experimental setup and the baseline models in Section 4.1. In Section 4.1, we construct an ocean-specific benchmark **OCEANBENCH** to evaluate the capabilities of our **OCEANGPT**. We describe the automatic and human evaluation in Section 4.1.

### 4.1 Implementation Details and Baselines

For the pre-training stage, we pre-train our **OCEANGPT** based on the LLaMA-2 [Touvron *et al.*, 2023b] for seven days with six A800 Nvidia GPUs. For the instruction-tuning stage, we employ the LoRA method [Hu *et al.*, 2021] to fine-tune the pre-trained model and choose three baseline models for comparison. We use the chat version of LLaMA-2 (*Llama-2-7b-chat-hf*) , which is a generative language model optimized for dialogue use cases. We also use *Vicuna-1.5* [Chiang *et al.*, 2023], a chat model which fine-tunes LLaMA-2 on dataset collected from ShareGPT. We further use *ChatGLM2-6B*, the optimized version of GLM [Zeng *et al.*, 2023]. The detailed experimental settings are shown in Table 2 (Appendix A).

**OCEANBENCH.** To evaluate the capabilities of LLMs for oceanography tasks, we design a benchmark called **OCEANBENCH**. Our benchmark includes a total of 15 ocean-related tasks such as question-answering, extraction, and description. Our evaluation samples are automatically generated from the seed dataset and have undergone deduplication [1] and manual verification by experts.

For the quality control, we further sample part of data and ask domain experts to evaluate the quality (those disagreed cases or bad cases will be manually fixed by domain experts.). The distribution of our desigend **OCEANBENCH** and the detailed statistics can be found in Table 1 and Figure 11.

| Task | Num | Task | Num |
|------|-----|------|-----|
| Analysis | 674 | Classification | 895 |
| Judgment | 655 | Letter Writing | 359 |
| Open-ended Generation | 930 | Extraction | 1,078 |
| Recommendation | 1,089 | Description | 1,246 |
| Summary | 149 | Editing | 1,075 |
| Identification | 464 | Transformation | 401 |
| Question Answering | 1,230 | Others | 157 |
| Commonsense Reasoning | 1,024 | | |

Table 1: The detailed statistics of **OCEANBENCH**.

**Metrics.** For the task-level calculation, we compare the effectiveness of two models for each task. When one model performs better on the majority of test samples in a single task, it is considered to 'win'

---

[1] We also perform deduplication between the benchmark and our training dataset to avoid the data leakage in the training stage of OceanGPT. The detailed explanation about the similarity calculating deduplication method is in Appendix A.

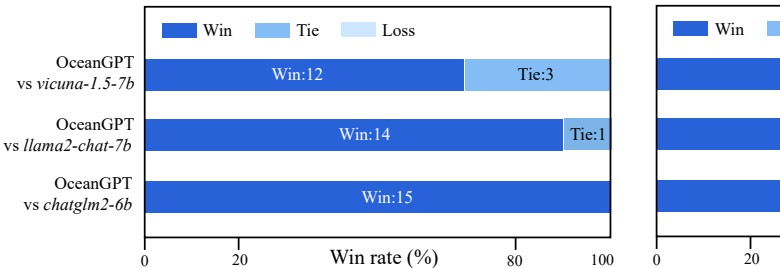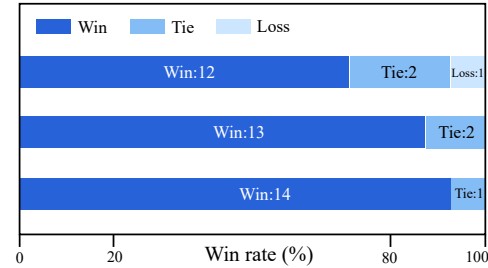

Figure 5: Ocean task-level results. **Left**: Automatic evaluation. **Right**: Human evaluation. Compared to baselines, **OCEANGPT** performs better than *llama2-chat-7b*, *vicuna-1.5-7b* and *chatglm2-6b* in both two settings. The instance-level result is in Figure 10 (Appendix A).

that task. For the instance-level computation process, we do not differentiate between specific tasks and instead calculate overall metrics.

**Automatic Evaluation.** To evaluate the performance and reduce reliance on manual evaluation, we leverage GPT-4 as the evaluator. Inspired by Wang *et al.* [2023c,b], we utilize an effective calibration method to evaluate the performance of two LLMs. For each testing question, we query the GPT4 to obtain the comparison result when given two outputs from two LLMs. We note that LLMs are sensitive to the position of responses, so alleviating the positional bias is very important. To balance the position bias, we exchange the order of the responses to form the new prompt. The final evaluating result is the sum of the test results for the two prompts with their order swapped.

**Human Evaluation.** To validate our proposed framework, we also collect the output data in different settings and evaluate it by human evaluation. We employ 5 students with an ocean science background as human annotators. For each evaluation setting, we sample a set of 200 examples and human annotators will rank the outputs they prefer. The total expense is about 500 US dollars.

## 5 Results

### 5.1 Insights from Performance Results

**OCEANGPT can obtain better performance than previous open-sourced LLMs.** In Figure 5, we compare the performance of **OCEANGPT** with the three baseline models across 15 sub-tasks at the task-level in the ocean domain. We utilize

both automatic and human evaluators, then compute the *win rate (%)* with baseline models. Compared to the baselines (*llama2-chat-7b*, *vicuna-1.5-7b*, *chatglm2-6b*)[2], **OCEANGPT** outperforms in the majority of tasks, which demonstrates the effectiveness of the proposed approach.

**OCEANGPT excels in a range of ocean science tasks.** As shown in Figure 6, we present detailed automatic evaluation experimental results in the **OCEANBENCH**. It can be clearly seen that our model is superior to baseline language models in the vast majority of tasks. Note that previous open-sourced LLMs even fail to handle several expertise ocean tasks (e.g., Editing). While our designed multi-agent data generation framework can effectively act as experts in various subfields of the ocean domain, which indicates that **OCEANGPT** is a better expert in various ocean domains.

**DOINSTRUCT are the effective ocean data generators by multi-agent collaboration.** As shown in Figure 7, we design three indicators to measure the data generation effect of our proposed method from the perspectives of **knowledge quality, expertise and diversity**. We use manual evaluation to calculate the scores of the three indicators from 1 to 5. The higher the score, the better the effect of the testing model. It can be seen that the evolving generator agent can effectively enhance the richness of ocean data. When the extraction agent is at work, the expertise of the content is greatly improved. At the same time, the inspector agent plays a significant role in enhancing the quality of the generated

---

[2]We have trained OceanGPT-7B, thus we only compare open-sourced LLMs around 7B.

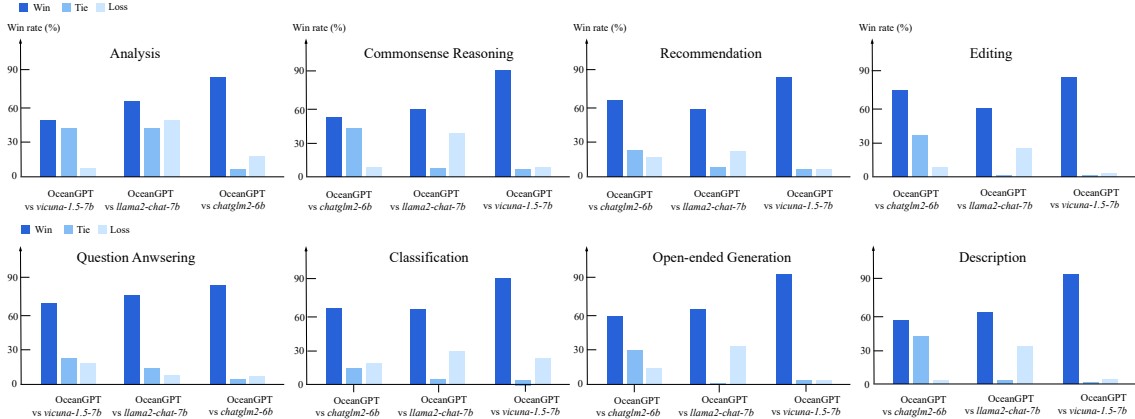

Figure 6: Evaluation results of **OCEANGPT** in the ocean science tasks in **OCEANBENCH**. The complete experimental results are shown in Figure 12 (Appendix A).

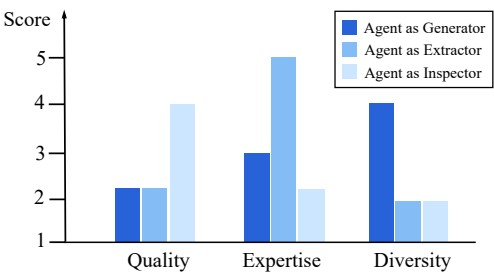

Figure 7: Performance analysis for different agents. We design three indicators to measure the generation effect.

data. It shows that multi-agent collaboration is effective for ocean instruction generation.

## 5.2 Exploring the Potential of OceanGPT

In this section, we explore the potential of **OCEANGPT** from the perspectives of ocean science and ocean engineering. For ocean science (Section 5.2), we focus on the key scientific issues of nuclear pollution in the ocean environment. For ocean engineering (Section 5.2), we explore the potential in robotics applications [Li *et al.*, 2023]. Specifically, we use Gazebo [3] as the simulator [Manhães *et al.*, 2016] to test **OCEANGPT**'s ability to control underwater robots.

**OceanGPT for Ocean Science**
In Figure 8, we compare the outputs of **OCEANGPT** and *vicuna-1.5-7b*. It shows

---

[3]https://github.com/uuvsimulator/uuv_simulator

that **OCEANGPT** shows a higher level of knowledge expertise when describing the content of radioactive nuclide research. Its textual content is not only clear in structure and well-organized, but also covers various aspects of radioactive nuclide research, from experimental design to data analysis, and then to risk assessment and disposal guidelines. In contrast, although *vicuna-1.5-7b* has clear expression and logicality, it lacks depth and specific content related to radioactive nuclides. Overall, **OCEANGPT** has advantages in terms of knowledge expertise, quality, and richness. The complete outputs are shown in the Table 6.

**OceanGPT for Ocean Engineering**
Ocean engineering focuses on the design, development, and management of structures and systems within the ocean environment. It plays an indispensable role in harnessing the vast potential of the oceans while ensuring sustainable and secure maritime operations. To facilitate interaction between **OCEANGPT** and the external world, we synthesize robotic code data and integrate those machine code instructions into the training data.

As depicted in Figure 9, **OCEANGPT** can instruct underwater robots via code or console commands, allowing them to execute basic path-finding operations. In this example, by using programming language as a prompt, our **OCEANGPT** can automatically generate code (the robot generate a double helix path) for underwater robot to operate complex tasks (based on human instructions). In fact, the ex-

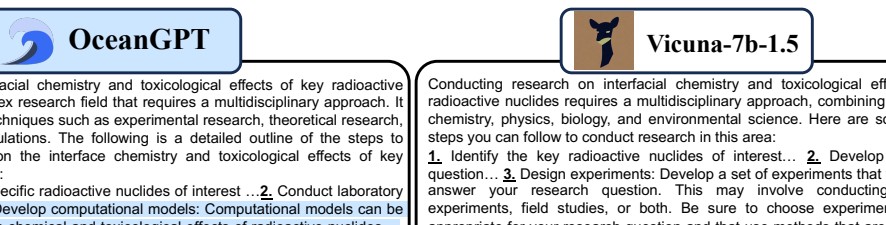

**OceanGPT**

The study of interfacial chemistry and toxicological effects of key radioactive nuclides is a complex research field that requires a multidisciplinary approach. It involves multiple techniques such as experimental research, theoretical research, and numerical simulations. The following is a detailed outline of the steps to conduct research on the interface chemistry and toxicological effects of key radioactive nuclides:

**1.** Determine the specific radioactive nuclides of interest …**2.** Conduct laboratory experiments … **3.** Develop computational models: Computational models can be used to simulate the chemical and toxicological effects of radioactive nuclides. … **4.**Analyze experimental data…**5.**Develop toxicological risk assessment methods: Toxicological risk assessment methods are used to evaluate the potential hazards of radioactive nuclides. This involves developing models that can predict the toxicological effects of radioactive nuclides and assessing their risks to human health. **6.** Conduct field experiments: ...7. Develop guidelines for safe disposal: Safe disposal of radioactive nuclides is critical for preventing environmental contamination and protecting human health. This involves...

**Vicuna-7b-1.5**

Conducting research on interfacial chemistry and toxicological effects of key radioactive nuclides requires a multidisciplinary approach, combining expertise in chemistry, physics, biology, and environmental science. Here are some general steps you can follow to conduct research in this area:

**1.** Identify the key radioactive nuclides of interest… **2.** Develop a research question… **3.** Design experiments: Develop a set of experiments that will help you answer your research question. This may involve conducting laboratory experiments, field studies, or both. Be sure to choose experiments that are appropriate for your research question and that use methods that are reliable and valid. **4.** Collect and analyze data… **5.** Interpret results and draw conclusions: Use your data to draw conclusions about the interfacial chemistry and toxicological effects of the key radioactive nuclides. Be sure to consider any limitations of your study and to consider how your findings may be generalized to other contexts. **6.** Communicate your results: Finally, share your findings with others through publication in academic journals, presentations at conferences, and other means. Be sure to clearly communicate the importance of your work and its implications...

Figure 8: Case analysis on ocean science task. We use blue font to represent the difference and the instruction is: *How to conduct research on interfacial chemistry and toxicological effects of key radioactive nuclides?*

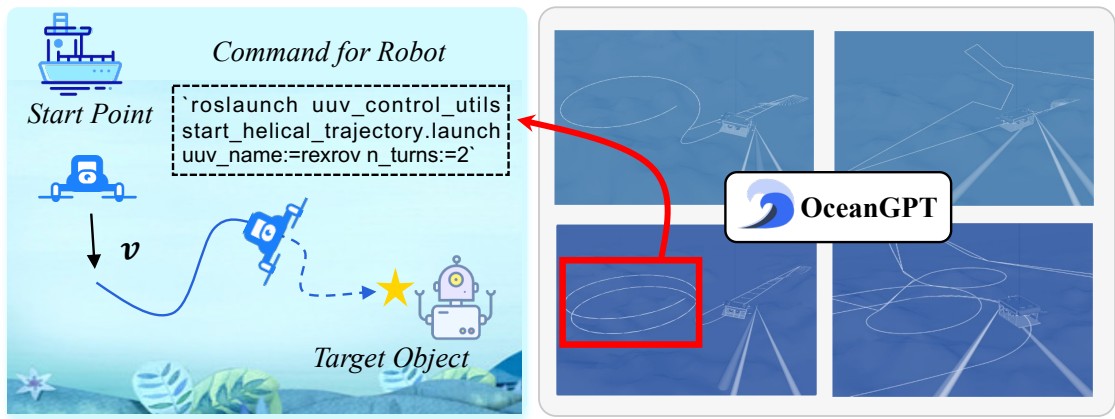

Figure 9: Our model can be instructed for underwater robot control in the simulation platform of Gazebo which shows OCEANGPT gains preliminary embodied intelligence capabilities.

perimental result suggests that **OCEANGPT** has the potential to acquire embodied intelligence. Though we make preliminary attempts for ocean robot interaction, it paves the way for advanced oceanic models to undertake intricate robotic control and complex planning tasks.

## 6   Conclusion

In this paper, we introduce **OCEANGPT**, the first-ever oceanographic pre-trained language model, which is expert in various ocean science tasks. To alleviate the difficulties for obtaining ocean data, we propose an domain construction framework called DOINSTRUCT, which constructs the ocean instruction dataset by multi-agent collaboration. Each agent in our designed framework is considered as an expert in a specific topic and is responsible for generating the corresponding data. Our generated dataset consists of diverse instructions to align the desired behaviors in ocean science issues. Additionally, we establish the first oceanography benchmark, **OCEANBENCH**, to evaluate the capabilities of LLMs in ocean domain. Though comprehensive analysis, we observe that **OCEANGPT** not only demonstrates a higher level of knowledge expertise for oceans science tasks but also gains preliminary embodied intelligence capabilities in ocean engineering. We will continue to improve **OCEANGPT** by training on larger corpus with larger models (e.g., 30B, 70B) and maintain **OCEANBENCH** by adding new data and tasks.

## Limitations

**Bias in Data Distribution**  In the realm of LLMs, the distribution of pre-training data and instruction data can be subject to substantial biases, which can shape the outputs of these models. Pre-training data for LLMs often comes from the internet, a vast and potentially biased source of information. The Internet content is inherently skewed, reflecting the biases of its contributors, and hence may not represent a balanced global perspective. Similarly, instruction data can also carry the biases of the humans who create these instructions. For instance, instruction developed by individuals with a particular cultural, socioeconomic, or educational background may inadvertently favor specific perspectives, languages, or communication styles and marginalize others. This bias in data distribution can result in models that reinforce existing prejudices, lack cultural sensitivity, or fail to accurately understand and generate content in underrepresented languages or dialects.

**Hallucination in LLMs**  Although LLMs have shown tremendous success in general domains of NLP, there is a notable issue regarding their tendency to produce hallucinations. Hallucinations refer to instances where LLMs occasionally generate content that deviates from the user's input, contradicts previously generated context, or conflicts with established world knowledge. By developing strategies to address the issue of hallucination, LLMs can better align their outputs with user intent, preserve coherence within generated content, and enhance their overall utility in real-world applications.

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

# A  Appendix

| Hyperparameter | Setting |
|---|---|
| Fine-tuning method | LoRA |
| Batch Size | 512 |
| Device† | NVIDIA A800 |
| GPU number | 6 |
| Learning Rate (LR) | $1e-4$ |
| LoRA $r$ | 8 |
| LoRA $\alpha$ | 16 |
| LoRA Dropout | 0.05 |
| Epoch | 10 |

Table 2: Detailed experimental settings.

## The Cost for Fine-tuning GPT-3.5-Turbo

For fine-tuning GPT-3.5-turbo, we use the reference code provided by OpenAI to fine-tune their language model. Overall, during the actual debugging process, we train and test the model multiple times, spending a total of nearly 500 US dollars (with the number of high-quality training samples being around 2000). Each time we run the script to train the model, it takes several hours.

The training cost is 0.008 USD per 1K tokens, the input cost during use is 0.012 USD per 1K tokens, and the output cost is 0.016 USD per 1K tokens. Assuming our prompt's input and output for one conversation is 1000 tokens, and if we have 2000 training samples with actual testing on 10000 samples, our training cost would be approximately 16.8 USD. The usage cost of the model after fine-tuning is about 138.0 USD, making the total cost around 154.8 USD. Since we debugged multiple times in the actual process, the real expenditure is greater. Overall, the overall training cost is not high and is affordable.

## Comparison between Our Fine-tuning Method and the Prefix Prompts

In the paper, we define 5 marine science topics, but this is a very broad categorization. In reality, each major topic contains many subtopics. For example, the topic 'Ecology and Environment' includes subtopics like marine meteorology, marine pollution, and over a dozen others. Altogether, these subtopics amount to over 500. Each of these subtopics is relatively independent and very important. Concatenating them as a prefix to GPT-3.5-turbo would **exceed its maximum length** limit and the actual usage cost would also be significant. Therefore, we believe that fine-tuning GPT-3.5-turbo is a better choice. The prompt examples are shown in Table 3 and Table 4.

## The Similarity Calculating Method in the Deduplication Procedure

Because comparing pairs for similarity involves a significant number of calculations, we choose a simple and effective method to address this challenge. We primarily use hash detection to compare two samples. First, we pre-extract keywords from the question part of each sample and then combine them into a new string. For example, the keywords for a data sample might be 'advice', 'ocean', and 'nuclear leakage'. We then employ hash detection to compare the keywords of the two samples. This method can relatively accurately prevent data leakage during the training process. It's important to note that sometimes the extraction of keywords can lead to redundancy or repetition, so we sometimes process them multiple times. Additionally, we also randomly select some samples and use the GPT-3.5-turbo API for detection to check for any cases of incomplete processing.

Additionally, regarding the deduplication process between the benchmark and our training dataset, we remove only a hundred or two hundred samples from the training set in the actual experiment, which is not a large number.

---

**Instruction:** You are a helpful ocean assistant. You are to extract the question from the provided content.
**Input:** Raw sentences in the marine literature (*The instruction prompt will be concatenated with raw sentences about seawater resources* ).

---

**Output:**
*Answer:* Existing methods of seawater resource exploitation have many problems, such as causing soil erosion and environmental pollution. Therefore, we need to seek more sustainable development methods, including water conservation, wastewater recycling, and the development of new water resources.
*Question:* Please discuss your views on the current methods of developing seawater resources.

---

Table 3: The prompt for fine-tuning GPT-3.5-turbo.

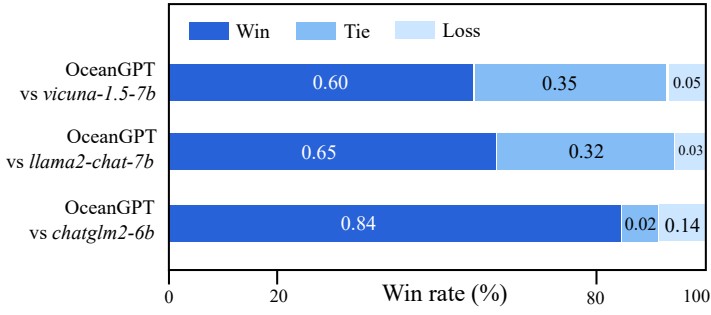

Figure 10: Instance-level results (automatic evaluation)

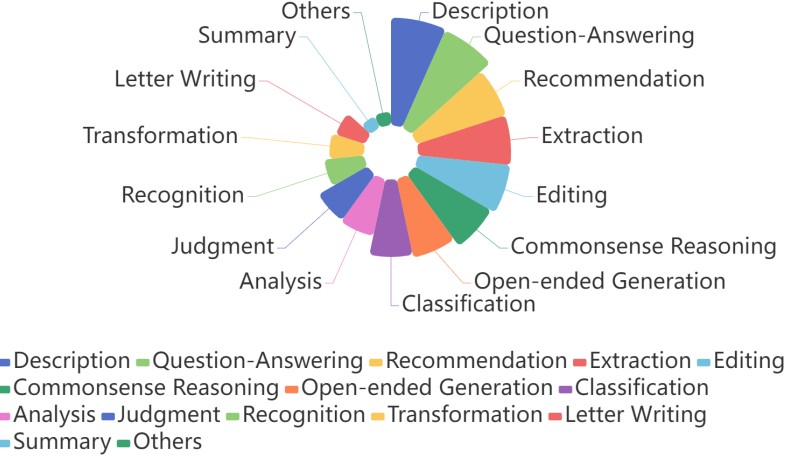

Figure 11: Distribution of our OCEANBENCH.

| Instruction: |
|---|
| You are a helpful ocean assistant. You are to extract the question from the provided content. |
| **Input:** |
| Raw sentences in the marine literature (*The instruction prompt will be concatenated with raw sentences about seawater resources* ). |

| The demonstration and answer pairs: |
|---|
| I will first give you some typical examples to help you become a marine expert. |
| Demonstration 1: ... Answer 1: ... |
| Demonstration 2: ... Answer 2: ... |
| Demonstration 3: ... Answer 3: ... |
| Demonstration 4: ... Answer 4: ... |
| ... |
| (*The demonstration and answer pairs for each marine subtopics. over 500 sub-categories. Each sub-categories has different task types* ) |

| Output: |
|---|
| *Answer: ... Question: ...* |
| (*Concatenating them as a prefix to GPT-3.5-turbo would exceed its maximum length limit and the actual usage cost is significant* ) |

Table 4: The prefix prompt to GPT-3.5-turbo.

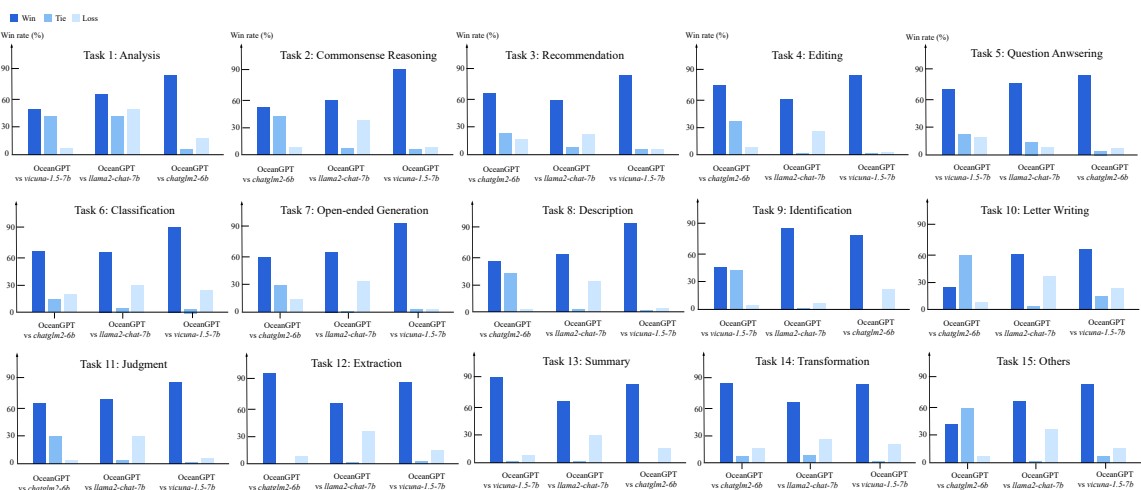

Figure 12: Automatic evaluation results of **OCEANGPT** in all tasks in **OCEANBENCH**.

**Prompt for "Fine-Tuned Agent as the Literature Extractor":**
You are a helpful ocean assistant.  You are to extract the question from each of the
answer provided.
Answer:  This is a seahorse, belonging to the family Syngnathidae.  Seahorses are
vertebrates commonly found in tropical and subtropical waters.  They have unique
morphology and biological characteristics and are important organisms in marine
ecosystems.

**Prompt for "Evolving Agent as the Generator":**
Assuming you are an expert in marine engineering and resources, please keep the
meaning of the following sentences unchanged and provide as much professional
knowledge as possible.
Sentences:Please recommend some mineral resources found in the East China Sea.

**Prompt for "Agent as the Inspector with Rule Constraints":**
Assuming you are an inspector in marine science, please filter and judge the sentences
in 'Sentences' based on the constraints provided below:
Constraints:  Keyword Filter:  Focus on literature that prominently mentions the
terms 'coral reefs', 'ocean acidification', or 'deep-sea exploration'.  Date Range:
Only consider articles published between 2010 and 2022.  Author Filter:  Prioritize
works by the Oceanic Research Institute.  Type of Literature:  Specifically look
for 'experimental studies' and 'review articles'.  Exclude 'conference papers'.
Geographical Focus:  Highlight research that pertains to the Pacific Ocean region.
Language Constraint:  Only select literature written in English.  Abstract Inclusion:
Ensure the abstract contains the phrase 'climate impact'.  Abstract Exclusion:
Exclude any literature whose abstract mentions 'laboratory simulation'.

**Prompt for automatic evaluation using GPT4:**
Please check if following sentences contain rich ocean related information.  If so,
output "related".  Otherwise, output "unrelated".
Sentences:  Dissolved organic carbon (DOC) represents the largest pool of reduced
carbon in oceans and plays important roles in the ocean carbon cycle and food webs .
DOC comprises nearly half of the riverine organic carbon flux into oceans.  Riverine
DOC is involved in numerous ecosystem functions, including key roles in chemical and
biological processes.  Refractory and labile DOC are, respectively, important for
carbon sequestra-tion in the ocean and a vital food source for marine bacteria.

Table 5: The prompt example that we use in this work.

| **Instruction:** How to conduct research on interfacial chemistry and toxicological effects of key radioactive nuclides? |
| --- |

| OCEANGPT | The study of interfacial chemistry and toxicological effects of key radioactive nuclides is a complex research field that requires a multidisciplinary approach. It involves multiple techniques such as experimental research, theoretical research, and numerical simulations. The following is a detailed outline of the steps to conduct research on the interface chemistry and toxicological effects of key radioactive nuclides: 1. Determine the specific radioactive nuclides of interest: Different radioactive nuclides have different chemical properties and toxicological effects. Therefore, it is necessary to select specific radioactive nuclides of interest and establish a research plan. 2. Conduct laboratory experiments: Laboratory experiments are crucial for studying the chemical properties and toxicological effects of radioactive nuclides. The experiments involve measuring the chemical properties of radioactive nuclides, such as their reactivity, stability, and chemical reactivity. 3. Develop computational models: Computational models can be used to simulate the chemical and toxicological effects of radioactive nuclides. This involves developing mathematical models and algorithms to predict the behavior of radioactive nuclides under different conditions. 4. Analyze experimental data: Experimental data obtained from laboratory experiments need to be analyzed and interpreted. This involves studying the experimental results and comparing them with theoretical predictions. 5. Develop toxicological risk assessment methods: Toxicological risk assessment methods are used to evaluate the potential hazards of radioactive nuclides. This involves developing models that can predict the toxicological effects of radioactive nuclides and assessing their risks to human health. 6. Conduct field experiments: Field experiments are necessary to study the behavior of radioactive nuclides in real-world environments. This involves measuring the chemical and toxicological effects of radioactive nuclides in different ecosystems and studying their distribution and transport. 7. Develop guidelines for safe disposal: Safe disposal of radioactive nuclides is critical for preventing environmental contamination and protecting human health. This involves developing guidelines for the safe disposal of radioactive nuclides and their storage and transport. |
| --- | --- |
| *vicuna-1.5-7b* | Conducting research on interfacial chemistry and toxicological effects of key radioactive nuclides requires a multidisciplinary approach, combining expertise in chemistry, physics, biology, and environmental science. Here are some general steps you can follow to conduct research in this area: 1. Identify the key radioactive nuclides of interest: Start by identifying the specific radioactive nuclides that you want to study. This may include elements such as uranium, plutonium, and cesium, among others. 2. Develop a research question: Once you have identified the key radioactive nuclides, develop a research question that addresses an important aspect of their interfacial chemistry and toxicological effects. This question should be specific and focused, and should guide your research efforts. 3. Design experiments: Develop a set of experiments that will help you answer your research question. This may involve conducting laboratory experiments, field studies, or both. Be sure to choose experiments that are appropriate for your research question and that use methods that are reliable and valid. 4. Collect and analyze data: Once you have conducted your experiments, collect and analyze the data you have generated. Be sure to carefully document your methods and results, and to interpret your data in a clear and concise manner. 5. Interpret results and draw conclusions: Use your data to draw conclusions about the interfacial chemistry and toxicological effects of the key radioactive nuclides. Be sure to consider any limitations of your study and to consider how your findings may be generalized to other contexts. 6.Communicate your results: Finally, share your findings with others through publication in academic journals, presentations at conferences, and other means. Be sure to clearly communicate the importance of your work and its implications for interfacial chemistry and toxicology. |

Table 6: Detailed case analysis on ocean science task. The input prompt is `How to conduct research on interfacial chemistry and toxicological effects of key radioactive nuclides?`

| Task Type | Example |
| --- | --- |
| Analysis | **Input:** Analyze the impact of marine environmental dynamics processes on the wave environment. **Output:** The dynamic processes of the oceanic environment have a significant influence on the wave conditions, such as the temperature of seawater, salinity, and ocean current affecting the height and frequency of waves. Meanwhile, the wave conditions can further impact the dynamics of ocean current and tides. |
| Classification | **Input:** Classify the ships in the East China Sea according to their type. **Output:** Ships in the East China Sea can be classified into categories such as merchant ships, fishing vessels, passenger ships, and warships, etc. Merchant ships are primarily used for cargo transport, fishing vessels for fishery resources, passenger ships for tourism and personnel transport, and warships for maritime security and defense. These diverse types of vessels are frequently active in the waters of the East China Sea. |
| Commonsense Reasoning | **Input:** Why are fish resources in the East China Sea gradually decreasing? **Output:** Fish resources in the East China Sea are gradually decreasing, mainly due to the impact of factors such as overfishing and environmental pollution. With the increase in population and economic development, fishing activities have been continuously strengthened, resulting in over-exploitation of fish resources. In addition, marine environmental pollution also has adverse effects on the survival and reproduction of fish resources. Therefore, protecting the marine environment and rationally developing fishing resources are key to ensuring the sustainable development of fish resources in the East China Sea. |
| Description | **Input:** Describe the classification and distribution areas of coral reefs. **Output:** Coral reefs are calcareous submarine structures formed by coral entities and their attachments. According to their crescent characteristics, they can be categorized into large ring coral reefs, small ring coral reefs, crisscross serrated coral reefs, irregular shaped coral reefs, etc. Approximately 70% of the world's coral reefs are distributed in tropical and subtropical sea areas, with the most important distribution areas being the western Indian Ocean, western Pacific Ocean, and eastern Pacific Ocean. |
| Editing | **Input:** Edit a big data report used for analyzing marine ecosystems. **Output:** Important parameters that can be tracked in marine ecosystems include water temperature, salinity, direction of waves, quantity and species of plankton, rate of ocean acidification, etc. In marine big data analysis, we use these data to train machine learning algorithms to understand the changes in the marine ecosystem, forecast potential environmental changes, etc. By utilizing these critical parameters, we can perceive the panorama and pattern of the marine ecosystem and extract valuable information from it to support adaptive and sustainable management practices in the ocean. |
| Extraction | **Input:** Extract the definition of the greenhouse effect and its causes. **Output:** The greenhouse effect refers to a thermal blanket formed by the absorption and reflection of solar heat radiation by greenhouse gases (such as carbon dioxide, methane, fluorocarbons, etc.) in the atmosphere, thereby raising the surface temperature of the earth. The sources of greenhouse gases include human activities, natural gases, deforestation, etc., which leads to an intensification of the greenhouse effect and global warming. |
| Judgment | **Input:** Is active intervention suitable for maintaining the coral reef ecosystem? **Output:** Active intervention can protect coral reef ecosystems, such as through artificial restoration, controlling sea water pollution, controlling carbon emissions, and reasonable development of coral reefs. Therefore, the coral reef ecosystem is suitable for active intervention and maintenance. |
| Letter Writing | **Input:** Write a letter to the local environmental protection agency about the waste discharge and pollution recently discovered in the sea area, proposing solutions. **Output:** Dear xx Environment Bureau: Recently, we have discovered a large amount of waste emission and pollution in the sea area, which has caused serious damage to our marine environment. We suggest that you take a series of measures to strengthen the monitoring and management of pollution sources in the sea area, punish illegal emissions in accordance with the law, and educate illegal polluting enterprises to enhance their environmental protection awareness. We hope that you can take action to effectively protect our marine environment. |
| Open-ended Generation | **Input:** Briefly introduce the principle and method of wave energy generation. **Output:** The principle of wave power generation is to convert wave energy into mechanical energy, and then turn the mechanical energy into electrical energy through a generator. The methods of wave power generation include: floating system, wave influx system, wave reflux system, and wave pressure system. |
| Question Answering | **Input:** What is three-dimensional digital ocean modeling technology? **Output:** Three-dimensional marine digital modeling technology involves the use of advanced remote sensing, lidar, sonar, and other detection technologies to acquire surface and subsurface oceanographic topographic data. Through digital modeling methods, a three-dimensional mathematical model of the ocean can be constructed. This type of model can provide information on topography, ocean currents, hydrology and acoustics, which is beneficial for applications in ocean resource development, environmental protection, and management, among others. |
| Recognition | **Input:** Identify the following sea fish species: sea bream, codfish, saury, and shark. **Output:** Dorada is a common type of sea fish, laterally compressed in form with a big mouth, and is known for its delicious meat. Cod is also a common sea fish, slender in shape with a soft hanging lower jaw, and is also a delicacy. Saury is a type of fish that enters ports from summer to autumn and can be made into a variety of dishes. Sharks are carnivorous marine animals with many species, some of which are primarily used as food, while others may attack humans. |
| Recommendation | **Input:** Recommend a desalination method suitable for deep-sea water resource development **Output:** The development of deep-sea water resources is suitable for the adoption of low-temperature seawater desalination technology, which uses cooling water to lower the seawater temperature below the ambient temperature, condensing out fresh water. Compared to traditional seawater desalination technology, low-temperature seawater desalination can conserve energy and reduce costs, while the resulting brine is also more concentrated. |
| Summary | **Input:** Provide a summary of the current status and future development prospects of offshore wind power generation technology. **Output:** Ocean wind power generation technology is one of the hot technologies in the field of renewable energy, its power generation efficiency is becoming increasingly high, equipment is becoming increasingly advanced, and the costs are gradually decreasing. It has a broad development prospect and can inject new development momentum into China's marine economy. |
| Transformation | **Input:** How to transform marine plants into green products? **Output:** The conversion process of marine plants can be divided into initial collection and extraction of effective parts, followed by processing to obtain the final product. Among them, extraction and processing are critical steps, and they vary for different marine plants. Common processing methods include alkali extraction, acid extraction, and enzyme extraction. The resulting green products also vary, such as chlorophyll, peptide substances, polysaccharide substances, etc. |

Table 7: Examples for tasks in **OCEANBENCH**.