# OpenReview forum: "OceanGPT: A Large Language Model for Ocean Science Tasks"
_ijcai.org/IJCAI/2024/Workshop/AI4Research — AI4Research 2024_

### Official Review · Reviewer_VMQ8 · 2024-06-01
**Evaluable reference for ocean science**

**Rating:** 7
**Confidence:** 3

**Review:**

Summary:
This research fine-tuned LLaMa 2 to OceanGPT using ocean science corpus so that OceanGPT can provide answers to oceanographic questions. The authors also proposed the DoINSTRUCT framework for building textural training data and OceanBench for model performance evaluation.


Strengths:
This study showcased the feasibility of using customized corpas to train LLMs. It could be a valuable reference for researchers in similar domains.

Weaknesses:
The adopted method is straightforward and commonly used for LLM fine-tuning; thus, the novelty is limited. In addition, the paper can provide necessary discussions and insights about the results and evaluation.

Comments:

1.	There is no support for the claim that OceanGPT gained “embodied intelligence capabilities”; this claim is vague.

2.	“Ocean data” in Line 54 is inaccurate; the authors may refer to only textual data rather than scientific observation data.

3.	Could the authors provide the comparison and discussion of the generated Gazebo code from OceanGPT and baseline models?

4.	Can the author briefly discuss some results from the GPT-3.5 and GPT-4? The readers may be interested in the results of the commercial models.

5.	Some terms can be spelled out, such as EHR and BM25, or brief introductions can be provided.

6.	When describing the agents in DoINSTRUCT, examples of input and output are recommended.

7.	Line 317, “it can automatically extract instructions (inst) from the unannotated ocean science corpus (output)”, does the “output” refer to the corpus? If so, it seems not correct.

8.	In the metric, what is the exact meaning of “performs better”? Does it mean more accuracy in fact/knowledge or language? Table 5 of the Appendix does not show the metric.

9.	The Limitations are too broad. They should be specific to this study.

---

### Official Review · Reviewer_f3c6 · 2024-06-02
**Great Advancement in Applying LLMs to Ocean Science**

**Rating:** 8
**Confidence:** 3

**Review:**

## Pros:

- The topic of applying AI to ocean science is popular and relevant to real-world problems like ocean engineering.
- The paper is well-written and organized.
- The proposed model outperforms other LLMs.
- The automated domain instruction framework, DoInstruct, is innovative and interesting.
- The OceanBench benchmark is of high quality.
- Extensive experiments compare OceanGPT with two different models.
- Both automatic and human evaluations are used, ensuring a robust evaluation process.

## Cons/Suggestions/Questions:

1. In the step where you mention "evolving agent as the Generator," agent (gpt-3.5-turbo) is asked to “refine the sample by conducting a more in-depth analysis of specific concepts or entities.” How do you ensure that these agents understand or have encountered these complex concepts or entities, which are written by annotators with rich backgrounds in ocean science? If the agent encounters a concept it does not understand, what is the course of action? Clarifying this could strengthen the DoInstruct framework.

2. For the automatic evaluation using GPT-4, position bias is considered. Did you identify any other biases? For example, does GPT-4 tend to prefer simpler or more direct answers, potentially lacking complexity? Conversely, does GPT-4 favor seemingly complex answers that do not adequately address the question? It would be beneficial to discuss whether you considered or addressed biases like these in evaluation.

3. Presentation issue: There is only one sub-section in section 4. Instead of using a leading paragraph and 4.1, you could directly start section 4.

---

### Decision · Program_Chairs · 2024-06-03

Accept